

# A transcriptomic study of probenecid on injured spinal cords in mice

Yu-Xin Zhang[1,2,3,*], Sai-Nan Wang[1,2,*], Jing Chen[1,2], Jian-Guo Hu[1,2] and He-Zuo Lü[1,2]

[1] Clinical Laboratory, The First Affiliated Hospital of Bengbu Medical College, Bengbu, China
[2] Anhui Key Laboratory of Tissue Transplantation, The First Affiliated Hospital of Bengbu Medical College, Bengbu, China
[3] Department of Biochemistry and Molecular Biology, Bengbu Medical College, Bengbu, China
* These authors contributed equally to this work.

Corresponding authors
Jian-Guo Hu, jghu9200@163.com
He-Zuo Lü, lhz233003@163.com

## ABSTRACT

**Background:** Recent studies have found that probenecid has neuroprotective and reparative effects on central nervous system injuries. However, its effect on genome-wide transcription in acute spinal cord injury (SCI) remains unknown. In the present study, RNA sequencing (RNA-Seq) is used to analyze the effect of probenecid on the local expression of gene transcription 8 h after spinal injury.
**Methods:** An Infinite Horizon impactor was used to perform contusive SCI in mice. The SCI model was made by using a rod (1.3 mm diameter) with a force of 50 Kdynes. Sham-operated mice only received a laminectomy without contusive injury. The injured mice were randomly assigned into either the control (SCI_C) or probenecid injection (SCI_P) group. In the latter group, the probenecid drug was intraperitoneally injected (0.5 mg/kg) immediately following injury. Eight hours after the injury or laminectomy, the spinal cords were removed from the mice in both groups. The total RNAs were extracted and purified for library preparation and transcriptome sequencing. Differential gene expressions (DEGs) of the three groups—sham, SCI_C and SCI_P—were analyzed using a DESeq software. Gene Ontology (GO) and Kyoto Encyclopedia of Genes and Genomes (KEGG) enrichment analysis of DEGs were performed using a GOseq R package and KOBAS software. Real-time quantitative reverse-transcriptase polymerase chain reaction was used to validate RNA-Seq results.
**Results:** RNA-Seq showed that, compared to the SCI_C group, the number of DEGs was 641 in the SCI_P group (286 upregulated and 355 downregulated). According to GO analysis, DEGs were most enriched in extracellular matrix (ECM), collagen trimer, protein bounding and sequence specific DNA binding. KEGG analysis showed that the most enriched pathways included: cell adhesion molecules, Leukocyte transendothelial migration, ECM-receptor interactions, PI3K-Akt signaling pathways, hematopoietic cell lineages, focal adhesions, the Rap1 signaling pathway, etc. The sequence data have been deposited into the Sequence Read Archive (https://www.ncbi.nlm.nih.gov/sra/PRJNA554464).

Subjects Bioinformatics, Genomics, Neuroscience, Neurology, Medical Genetics
Keywords Probenecid, Spinal cord injury, RNA sequencing, Mice

## INTRODUCTION

The term spinal cord injury (SCI) refers to a variety of injuries to the spinal cord. According to the severity of injury the symptoms may vary, ranging from pain to a complete loss of movement and sensory function. SCI affects millions of people worldwide, typically for life (*Friedli et al., 2015*). In the United States, there are around 12,000–20,000 new SCI cases a year, with more than 280,000 patients confined to wheelchairs (*Singh et al., 2014*). In the past decade, the SCI cases in China have risen tenfold, and are now increasing by about 60,000 cases every year (*Qiu, 2009*). SCI has a high rate of disability and mortality, which creates a heavy burden for patients, their families and society (*Krueger et al., 2013*). Therefore, it is imperative to explore effective treatment methods for repairing SCI in order to improve the quality of life of patients and reduce the burden of social medical care.

Pathological processes following traumatic SCI can be characterized as primary and secondary injuries (*Geisler et al., 2002*; *McDonald & Sadowsky, 2002*). Primary injury refers to the direct injury of the spinal cord by mechanical force, including compression, contusion, laceration and penetration. Secondary injury refers to edema, ischemia, local inflammation and electrolyte changes. These changes can cause an accumulation of lipid peroxides and oxygen-free radicals, as well as a release of inflammatory factors and proteases. This can ultimately lead to a large amount of cell apoptosis or necrosis, which further aggravates the damage to the neurons and axons (*Ahuja et al., 2017*; *Oyinbo, 2011*; *Tran, Warren & Silver, 2018*).

Probenecid is an organic anion-transport protein inhibitor, which has been widely used in clinical settings (*Hagos et al., 2017*; *Töllner et al., 2015*). For example, probenecid has been used as a synergist in the treatment of gout and antibiotics (*Baranova et al., 2004*; *Papadopoulos & Verkman, 2008*). It can reduce the degree of cognitive impairment in afflicted rats (*Mawhinney et al., 2011*), as well as reverse cerebral ischemic injury and cellular inflammation (*Wei et al., 2015*; *Xiong et al., 2014*). The combination of probenecid and N-Acetylcysteine could potentially both maintain intracellular GSH concentrations and inhibit neuronal death after a traumatic stretch injury (*Du et al., 2016*). Some studies report that probenecid can also reduce neuropathic pain in the spinal cord (*Bravo et al., 2014*; *Pineda-Farias et al., 2013*). Therefore, these reports indicate that probenecid has neuroprotective and reparative effects on central nervous system (CNS) injuries. However, whether the drug can play a role in treating SCI and whether it can affect the gene expression profiles in injured spinal cords remain unknown. To test this, probenecid was injected intraperitoneally into spinal cord-injured mice immediately after injury. Eight hours after the injury or laminectomy, the spinal cords were removed and RNA-Seq was used to analyze the changes in transcriptome expression in the injured area. Then, the key molecules and signal pathways were screened and identified, providing a new theoretical and experimental basis for SCI clinical treatment.

## MATERIALS AND METHODS

### Animals

A total of 27 healthy, clean C57BL/6 female mice (18–20 g, 8 weeks old) were used to model SCI. The Animal Care and Use Committee of Bengbu Medical College provided full

approval for this research (037/2017). Animal care following surgery was carried out in compliance with the regulations for the management of experimental animals (revised by the Ministry of Science and Technology of China in June 2004), as well as the guidelines and policies on rodent survival surgery provided by the Animal Care and Use Committee of Bengbu Medical College.

## Contusive SCI and drug injection

An Infinite Horizon impactor (Precision Systems & Instrumentation, Lexington, KY, USA) was used to perform contusive SCI on the mice. These mice were first anesthetized with 50 mg/kg pentobarbital, followed by the excision of the T9 lamina. A SCI model of this procedure was created using a rod (1.3 mm diameter) with a force of 50 Kdynes. Sham-operated (sham) mice only received a laminectomy without contusive injury.

The spinal cord-injured mice were randomly assigned to the solvent control (SCI_C) or probenecid injection (SCI_P) group. The solvent or probenecid (0.5 mg/kg) was intraperitoneally injected immediately following injury. The solution (pH 7.3) was prepared as previously described (*Hainz et al., 2017*).

## RNA isolation, quantification and qualification

Eight hours after the injury or laminectomy, the mice were anesthetized and perfused with 10 ml PBS. Their spinal cords (0.5 cm including the injury center) were then removed. The total RNAs from their spinal cords were extracted and purified as previously described (*Shi et al., 2017*).

## Library preparation and transcriptome sequencing

The sequencing libraries were produced by using a NEBNext® Ultra™ RNA Library Prep Kit for Illumina® (New England Biolabs, Ipswich, MA, USA) as previously described (*Shi et al., 2017*). Finally, the 125 bp/150 bp paired-end reads were obtained and sequenced on an Illumina Hiseq platform.

## Analysis of differentially expressed gene

Prior to DEG analysis, the gene expression statistics were analyzed using RSEM software (http://deweylab.github.io/RSEM/) to convert the read count numbers to Fragments Per Kilobase of transcript per Million fragments mapped (FPKM), and Principal Component Analysis (PCA) analysis was conducted to determine the similarities and differences in the data. DEGs of the three groups of mice were analyzed as previously described (*Shi et al., 2017*) using DESeq software (http://www.bioconductor.org/). Benjamini and Hochberg's approach was used to control the false discovery rate and adjust the *P*-values. The adjusted *P*-value < 0.05 was defined as a standard for significant differences in gene expression. In addition to FPKM hierarchical clustering analysis of DEGs, we further analyzed the subclusters based on log2 (ratios) of their gene expression level relative to that of sham group. The log2 (ratios) in the SCI_C group ≥1 or ≤−1 was used as a cutoff for subcluster analysis. The clustering algorithm divided the DEGs with similar gene expression trends into several subclusters.

**Table 1 PCR primers used in the study.**

| Gene | Forward primer 5′–3′ | Reverse primer 5′–3′ |
|---|---|---|
| Itga1 | TCAGTGGAGAGCAGATCGGA | CCTCGTCTGATTCACAGCGT |
| lamb1 | TGCCTTTTCTCCCCGCTACC | CCATGTCCAGTCCTCGCAGA |
| Cldn5 | TTCTATGATCCGACGGTGCC | CTTGACCGGGAAGCTGAACT |
| CD34 | ACCACAGACTTCCCCAACTG | CATATGGCTCGGTGGGTGAT |
| lama2 | GCATTAGTGAGCCGCCCTAT | TCTTTCAGGTCTCGTGTGGC |
| Esam | AGACTCTGGGACTTACCGCT | GGTCACATTGGTCCCGACAT |
| Setdb2 | CCACAAATGGAGATCATACACCT | GCAGTGGGGCTTCCTTTTTC |
| Agrn | CTCTGCCACTGGAACACAGA | GGAAAAGCAGCACCGCAAAG |
| Ccnt2 | AGCAAGGATTTGGCACAGAC | CTCTAGGGTAACCGTGGGGT |
| beta-actin | AGAAGCTGTGCTATGTTGCTCTA | ACCCAAGAAGGAAGGCTGGAAAA |

## Gene ontology and kyoto encyclopedia of genes and genomes enrichment analysis of DEGs

The GO and KEGG analysis were performed using a GOseq R package and KOBAS software as previously described (*Shi et al., 2017*). In GO analysis, DEGs were implemented using the GOseq R package and gene length bias was corrected. GO terms with corrected *P* value ≤ 0.05 were considered significantly enriched by DEGs. KEGG is a database resource for understanding the high-level functions and utilities of the biological system (http://www.genome.jp/kegg/). In this study, we used KOBAS software to test the statistical enrichment of DEGs in KEGG pathways.

## Real-time quantitative reverse-transcriptase polymerase chain reaction

To validate RNA-Seq results, nine DEGs were randomly selected and verified via Real-time quantitative reverse-transcriptase polymerase chain reaction (RT-qPCR) according to our previous methods (*Shi et al., 2017*). The analysis was performed in six samples, which included three independent samples and duplicates of these samples to be used in RNA-seq analysis. PCR primer sequences are listed in Table 1. The relative quantitative results of each group of genes were calculated according to the formula "$^{\Delta\Delta}Ct$" (*Livak & Schmittgen, 2001*). The statistical values ($n = 6$/group) were presented as mean ± standard deviation (SD). The data were analyzed using one-way Analysis of Variance (ANOVA), followed by Student–Newman–Keuls tests. Statistical differences were considered significant at $P < 0.05$.

## RESULTS

### Identification of expressed transcripts the mice spinal cords

For the high quality assessment of sequencing data, nine cDNA libraries were established, including sham (sham_1, sham_2 and sham_3), SCI_C (SCI_C1, SCI_C2 and SCI_C3) and SCI_P (SCI_P1, SCI_P2 and SCI_P3). RNA-Seq produced 48,848,744–61,037,096 raw reads for each sample. After filtering out the low-quality reads, there were 48,226,002–60,037,772 clean reads, with the Q30 (%) 93.67–94.31 (Table 2).

**Table 2 Summary of sequence assembly after Illumina sequencing.**

| Sample name | Raw reads | Clean reads | clean bases | Error rate (%) | Q20 (%) | Q30 (%) | GC content (%) |
|---|---|---|---|---|---|---|---|
| Sham_1 | 56,509,230 | 55,796,658 | 8.37G | 0.03 | 97.73 | 93.95 | 51.23 |
| Sham_2 | 48,848,744 | 48,226,002 | 7.23G | 0.03 | 97.6 | 93.67 | 51.71 |
| Sham_3 | 58228350 | 57,459,748 | 8.62G | 0.03 | 97.67 | 93.78 | 51.42 |
| SCI_C1 | 58,862,872 | 58,126,844 | 8.72G | 0.03 | 97.88 | 94.31 | 51.39 |
| SCI_C2 | 56,980,070 | 56,166,058 | 8.42G | 0.03 | 97.74 | 94.03 | 51.42 |
| SCI_C3 | 59,804,518 | 58,798,224 | 8.82G | 0.03 | 97.63 | 93.74 | 51.02 |
| SCI_P1 | 54,853,344 | 53,996,254 | 8.1G | 0.03 | 97.72 | 93.91 | 50.93 |
| SCI_P2 | 56,322,736 | 55,540,308 | 8.33G | 0.03 | 97.87 | 94.27 | 50.94 |
| SCI_P3 | 61,037,096 | 60,037,772 | 9.01G | 0.03 | 97.71 | 93.89 | 50.92 |

Note:
Sham: Sham_1, Sham_2, Sham_3; SCI (solvent control): SCI_C1, SCI_C2, SCI_C3; SCI (probenecid): SCI_P1, SCI_P2, SCI_P3; Q20: the percentage of bases with a Phred value >20; Q30: the percentage of bases with a Phred value >30.

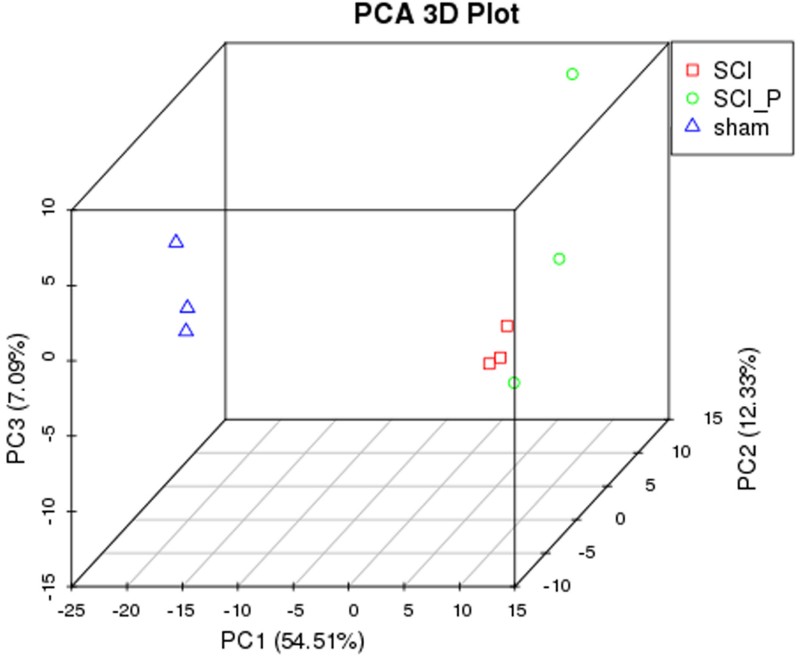

**Figure 1 PCA analysis.** PCA analysis was performed using three principal components (PC1, 2, and 3) to demonstrate the source of variance ($n = 3$).

In order to identify the source of variation within the original data, PCA analysis was conducted. As shown in Fig. 1, PC1, PC2 and PC3 were 54.51%, 12.33% and 7.09%, respectively. Although not too far from one another, the distance between SCI_C (or SCI_P) and sham was apparent and sufficient for the analysis. These distances demonstrated that the data could be used for the next analysis.

### Effect of SCI and probenecid treatment on gene expression

RPKM and DEGSeq were used to analyze the gene expression level and differential expression profiles, respectively. The results showed that, as compared to the sham group,
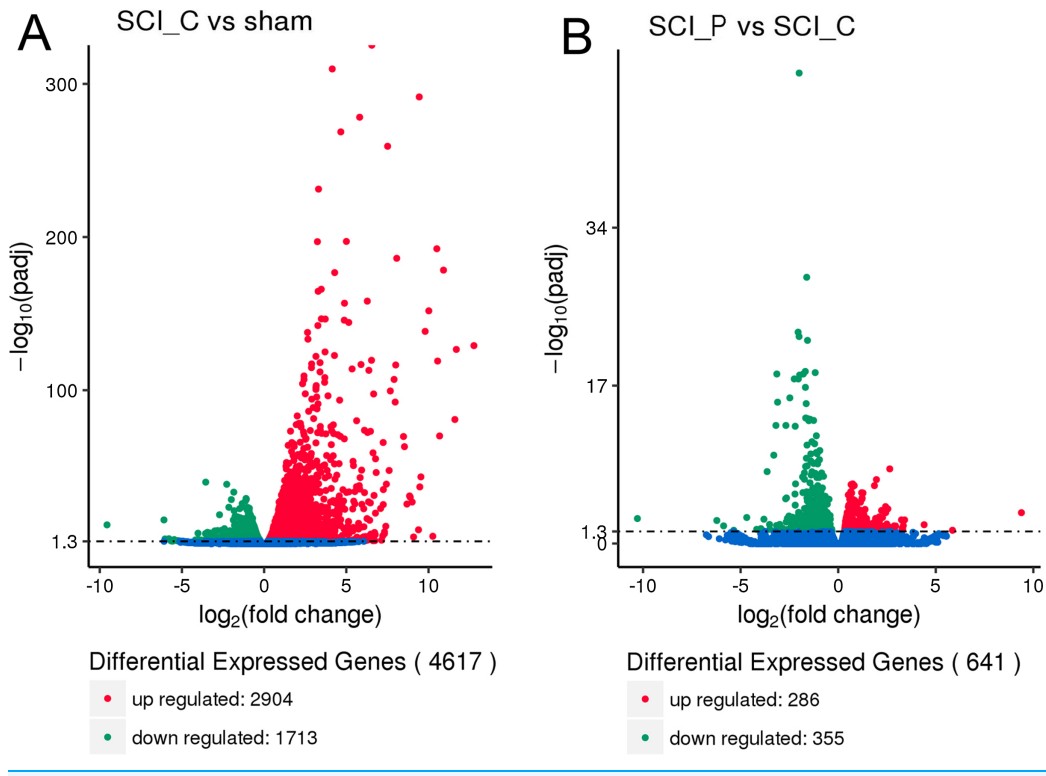

Figure 2 **Volcano map of DEGs.** Red, green and blue dots represent significantly upregulated, downregulated and no changed gene expressions, respectively. (A) SCI_C vs. Sham and (B) SCI_P vs. SCI_C.

there were 4,617 DEGs in the SCI_C group, including 2,904 upregulated and 1,713 downregulated genes (Fig. 2A; Table S1). Compared to the SCI_C group, there were 641 different genes in the SCI_P group, 286 were upregulated and 355 were downregulated (Fig. 2B; Table S1). The sequence data have been deposited into Sequence Read Archive (https://www.ncbi.nlm.nih.gov/sra/PRJNA554464).

### RT-qPCR identification of DEGs

In order to verify the RNA-Seq results, nine DEGs were randomly selected from the SCI_P group, as compared with the SCI_C group, namely Itga1, Lamb1, Cldn5, Lama2, CD34, Esam, Setdb2, Agrn and Ccnt2. The RNA-Seq and RT-qPCR results indicated that the expression patterns of these DEGs were similar (Fig. 3).

### Cluster analysis of DEGs

The DEGs from different groups were analyzed using FPKM hierarchical cluster analysis. As shown in Fig. 4, DEGs were hierarchically clustered and classified into different expression clusters. These clusters contained upregulated or downregulated DEGs. When compared to the sham group, most upregulated DEGs in the SCI_C group were in the middle and upper clusters, while downregulated DEGs were delegated to the lower cluster. Additionally, compared to the sham group, most upregulated DEGs in the SCI_P group were in the upper cluster, while downregulated DEGs were mainly observed in the

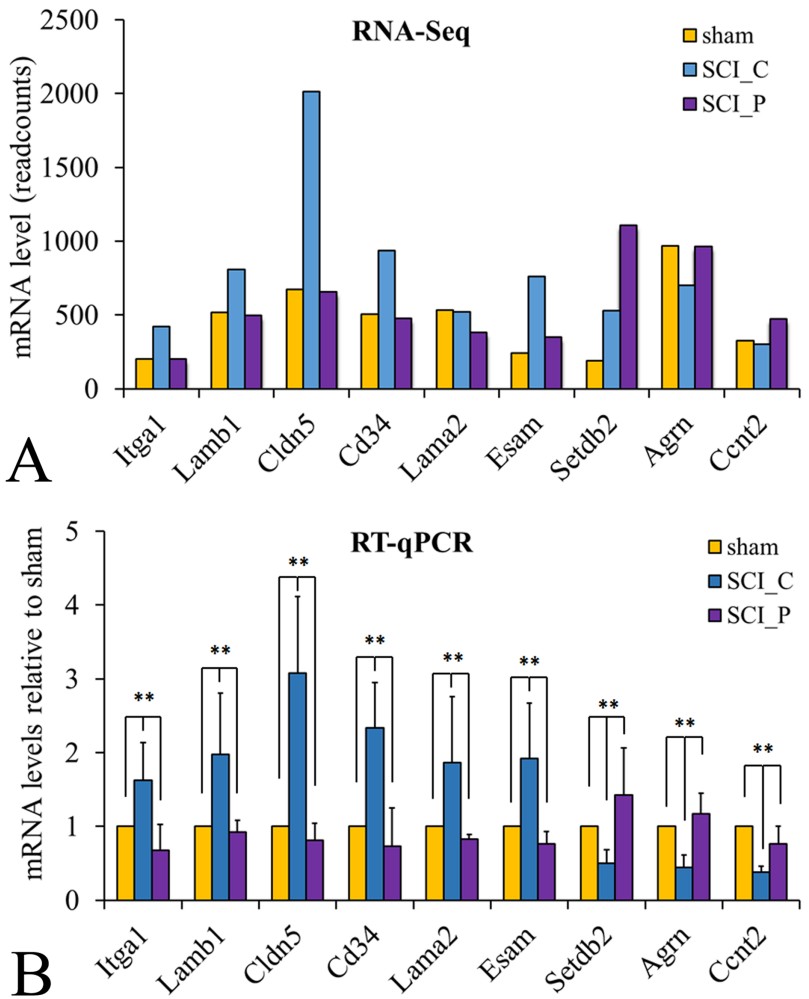

**Figure 3  RT-qPCR verification of DEGs characterized by RNA-Seq.** (A) The longitudinal coordinates in RNA-Seq were the mRNA expression level (read counts, $n = 3$). (B) The longitudinal coordinates in RT-qPCR were the mRNA expression level calculated using the $^{\Delta\Delta}$Ct method and expressed relative to the value in the sham group (designated as oness). All data were calculated with mean ± standard deviation ($n = 6$, which included three independent samples and the three same samples used for the RNA-seq analysis). **$P < 0.01$ (ANOVA).    

lower cluster. When compared to the SCI_C group, some upregulated DEGs in the SCI_P group were observed in upper cluster, while downregulated DEGs were observed in the middle cluster (there were also some clusters in this grouping that showed no significant differences).

In addition to FPKM hierarchical clustering analysis of DEGs, the subclusters—which have similar expression trends—were further analyzed. The log2 (ratios) in the SCI_C group ≥1 or ≤−1 was used as a cutoff for subcluster analysis. As shown in Fig. 5, we found several subclusters with similar expression trends. Based on log2 (ratios) of their gene expression levels relative to that of the sham group, the log2 (ratios) of all gene expression levels in the sham group were zero. Figures 5A and 5B show that the two subclusters were strongly upregulated following SCI and then downregulated upon probenecid treatment.
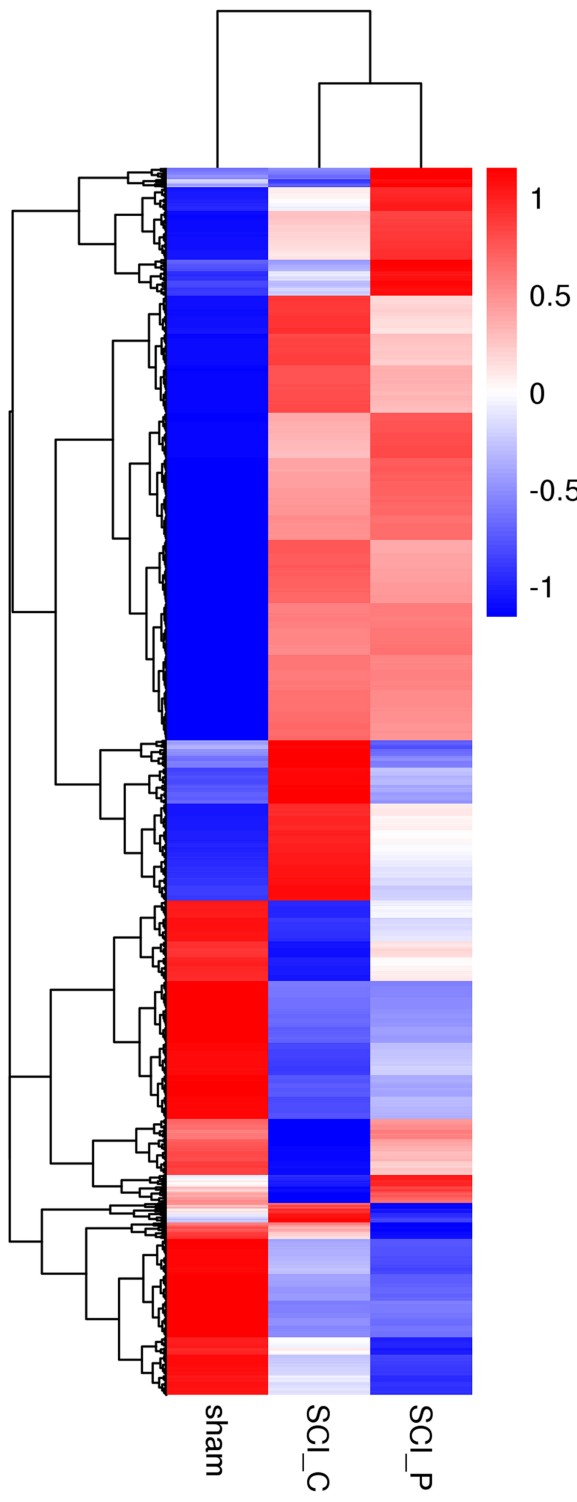

**Figure 4 Hierarchical cluster analysis of DEGs.** The DEGs in different groups were analyzed using FPKM hierarchical cluster analysis. The read count numbers of FPKM were converted by RSEM software. DEGs were classified into different expression cluster by hierarchical clustering. The color scheme (red to blue) represents the up to down of the gene expression. sham: sham group; SCI_C: SCI (solvent control) group; SCI_P: SCI (probenecid) group.

**A**

| Name | sham | SCI_C | SCI_P |
| --- | --- | --- | --- |
| Rgs16 | 0 | 1.78 | 1.07 |
| Col4a1 | 0 | 1.7 | 0.9 |
| Apobec3 | 0 | 1.66 | 1.29 |
| Syk | 0 | 1.65 | 0.89 |
| Akr1b8 | 0 | 1.64 | 1.18 |
| Ifi35 | 0 | 1.62 | 1.21 |
| Thbs4 | 0 | 1.61 | 1.2 |
| Esam | 0 | 1.61 | 0.49 |
| Pstpip2 | 0 | 1.59 | 0.58 |
| Sdc1 | 0 | 1.58 | 1.05 |
| Gda | 0 | 1.58 | 0.85 |
| A2m | 0 | 1.57 | 1.42 |
| Dok2 | 0 | 1.57 | 0.66 |
| Nupr1 | 0 | 1.57 | 0.85 |
| Aoc3 | 0 | 1.57 | -0.41 |
| Zfp429 | 0 | 1.54 | 0.65 |
| Loxl4 | 0 | 1.53 | 1.05 |
| Capg | 0 | 1.52 | 0.9 |
| Tnfrsf11b | 0 | 1.52 | 1.37 |
| Timeless | 0 | 1.51 | 1.09 |
| Plek | 0 | 1.51 | 0.96 |
| Naip2 | 0 | 1.51 | 1.22 |
| Arhgef5 | 0 | 1.49 | 0.59 |
| Slc1a5 | 0 | 1.48 | 0.68 |
| Plekhg2 | 0 | 1.48 | 0.83 |
| Anxa2 | 0 | 1.47 | 1.05 |
| Cyp1b1 | 0 | 1.47 | 0.85 |
| Tbx3 | 0 | 1.46 | 0.29 |
| Tmem154 | 0 | 1.45 | 1 |
| Emp3 | 0 | 1.44 | 1.12 |
| Ptprc | 0 | 1.43 | 0.57 |
| Ets1 | 0 | 1.42 | 0.68 |
| Mgp | 0 | 1.41 | 0.28 |
| Il33 | 0 | 1.41 | 0.85 |
| Acat3 | 0 | 1.41 | 0.98 |
| Anxa3 | 0 | 1.4 | 0.97 |
| Cd52 | 0 | 1.4 | 0.22 |
| Ltbr | 0 | 1.4 | 1.06 |
| Gjc1 | 0 | 1.4 | 0.87 |
| Igfbp7 | 0 | 1.39 | 0.47 |
| Stom | 0 | 1.39 | 0.91 |
| Gpr182 | 0 | 1.39 | 0.46 |
| Hmga2 | 0 | 1.39 | 0.81 |
| Steap3 | 0 | 1.39 | 0.73 |
| Ncf2 | 0 | 1.37 | 0.96 |
| Plp2 | 0 | 1.37 | 0.77 |
| Phf11d | 0 | 1.37 | 1.09 |
| Nos3 | 0 | 1.35 | 0.06 |
| Hspa4l | 0 | 1.35 | 0.79 |
| Adam12 | 0 | 1.34 | 0.82 |
| Tgfb1 | 0 | 1.34 | 0.57 |
| Rdh12 | 0 | 1.33 | 0.66 |
| Marveld2 | 0 | 1.33 | 0.91 |
| Alox5ap | 0 | 1.3 | 0.03 |
| Ap1s3 | 0 | 1.29 | 1.03 |
| Ampd1 | 0 | 1.29 | -1.8 |
| Itgb2 | 0 | 1.29 | 0.33 |
| Ip6k3 | 0 | 1.28 | 1.04 |
| Snx20 | 0 | 1.28 | 0.5 |
| Flt4 | 0 | 1.27 | 0.31 |
| Spi1 | 0 | 1.26 | 0.71 |
| Fblim1 | 0 | 1.24 | 0.81 |
| Filip1l | 0 | 1.24 | 0.41 |
| Was | 0 | 1.22 | 0.99 |
| Arhgap30 | 0 | 1.22 | 0.83 |
| Slc39a1 | 0 | 1.21 | 0.96 |
| Rin3 | 0 | 1.21 | 0.39 |
| Emilin1 | 0 | 1.21 | 0.7 |
| Erg | 0 | 1.21 | 0.13 |
| Entpd1 | 0 | 1.2 | 0.46 |
| Notch4 | 0 | 1.2 | 0.43 |
| Pear1 | 0 | 1.19 | 0.08 |
| Hcls1 | 0 | 1.18 | 0.76 |
| Nfam1 | 0 | 1.17 | 0.49 |
| Parp10 | 0 | 1.16 | 0.81 |
| Dkk2 | 0 | 1.15 | -0.12 |
| Zfp217 | 0 | 1.15 | 0.79 |
| Iqgap1 | 0 | 1.14 | 0.64 |
| Ier5l | 0 | 1.14 | 0.71 |
| Msn | 0 | 1.13 | 0.73 |
| Icam2 | 0 | 1.13 | 0.1 |
| Itgam | 0 | 1.13 | 0.57 |
| Plekha4 | 0 | 1.12 | -0.33 |
| Irf8 | 0 | 1.12 | 0.25 |
| C1ra | 0 | 1.12 | 0.47 |
| Atp10a | 0 | 1.11 | 0.51 |
| Slc25a24 | 0 | 1.1 | 0.74 |
| Cd33 | 0 | 1.1 | 0.68 |
| Tarm1 | 0 | 1.1 | 0.24 |
| Kcnj8 | 0 | 1.1 | 0.38 |
| Hmgcs2 | 0 | 1.09 | -0.61 |
| Trim56 | 0 | 1.07 | 0.73 |
| Sp100 | 0 | 1.07 | -0.1 |
| Trpm6 | 0 | 1.07 | 0.39 |
| Erbb2 | 0 | 1.06 | 0.45 |
| Id3 | 0 | 1.06 | 0.51 |
| Tbxa2r | 0 | 1.05 | -0.96 |
| Foxq1 | 0 | 1.05 | -0.44 |
| Myo1c | 0 | 1.05 | 0.6 |
| Arhgdib | 0 | 1.05 | 0.34 |
| Fbln2 | 0 | 1.04 | 0.3 |
| Apobr | 0 | 1.04 | 0.37 |
| Hk3 | 0 | 1.04 | 0.5 |
| Fxyd3 | 0 | 1.02 | -0.32 |
| Cybb | 0 | 1 | -0.71 |
| Ptrh1 | 0 | 1 | 0.33 |

**B**

| Name | sham | SCI_C | SCI_P |
| --- | --- | --- | --- |
| Sox7 | 0 | 1.49 | 0.65 |
| S100a6 | 0 | 1.48 | 1.34 |
| Tnfrsf10b | 0 | 1.48 | 0.96 |
| Lyn | 0 | 1.42 | 1.2 |
| Trib3 | 0 | 1.4 | 1.03 |
| Tpm4 | 0 | 1.39 | 1.02 |
| Rac2 | 0 | 1.36 | 1.06 |
| Tec | 0 | 1.3 | 1.15 |
| Wwtr1 | 0 | 1.29 | 1.03 |
| Slc5a3 | 0 | 1.27 | 1.09 |
| Yap1 | 0 | 1.27 | 0.98 |
| Fcer1g | 0 | 1.26 | 1.06 |
| Ecm1 | 0 | 1.25 | 0.61 |
| Ptpn12 | 0 | 1.25 | 1.08 |
| Itpripl1 | 0 | 1.23 | 0.86 |
| Gpd1 | 0 | 1.22 | 1.1 |
| Id1 | 0 | 1.2 | 0.62 |
| S100a10 | 0 | 1.19 | 1 |
| Met | 0 | 1.18 | 1.01 |
| Wisp1 | 0 | 1.18 | 1.01 |
| Slc2a1 | 0 | 1.18 | 0.72 |
| Twist1 | 0 | 1.17 | 0.26 |
| Mb21d1 | 0 | 1.17 | 0.88 |
| Ddx58 | 0 | 1.16 | 1.02 |
| Layn | 0 | 1.16 | 1.12 |
| Tmem37 | 0 | 1.16 | 0.51 |
| Cavin1 | 0 | 1.15 | 0.65 |
| Ldha | 0 | 1.14 | 0.74 |
| Lrrn4cl | 0 | 1.14 | 0.28 |
| Cyba | 0 | 1.14 | 0.56 |
| Adipor2 | 0 | 1.13 | 0.97 |
| Myh9 | 0 | 1.12 | 0.79 |
| Casp12 | 0 | 1.12 | 0.99 |
| Vsig2 | 0 | 1.11 | 0.4 |
| Fhl3 | 0 | 1.11 | 0.9 |
| Rhoc | 0 | 1.11 | 0.91 |
| Rbpms | 0 | 1.1 | 0.29 |
| Lrrc8a | 0 | 1.1 | 0.98 |
| Xbp1 | 0 | 1.1 | 0.64 |
| Susd6 | 0 | 1.09 | 0.98 |
| Cdc42se1 | 0 | 1.07 | 1 |
| Myo1g | 0 | 1.07 | 0.69 |
| Rph3al | 0 | 1.07 | 0.72 |
| Nfya | 0 | 1.06 | -0.11 |
| Cflar | 0 | 1.05 | 0.67 |
| Psmb8 | 0 | 1.04 | 0.7 |
| Vgf | 0 | 1.04 | 0.69 |
| Dll4 | 0 | 1.03 | 0.42 |
| Tnfaip8l1 | 0 | 1.03 | 0.84 |
| Ncf1 | 0 | 1.03 | 0.77 |
| Gypc | 0 | 1.02 | 0.74 |
| Cd63 | 0 | 1.02 | 0.86 |
| Psd4 | 0 | 1.01 | 0.72 |
| Tspo | 0 | 1 | 0.56 |

**C**

| Name | sham | SCI_C | SCI_P |
| --- | --- | --- | --- |
| Snrnp40 | 0 | -2.24 | -1.44 |
| Ranbp3l | 0 | -2.18 | -1.9 |
| Wdr49 | 0 | -2.11 | -0.93 |
| Ly6g6f | 0 | -1.86 | -1.72 |
| Lrrc43 | 0 | -1.68 | -1.49 |
| Gpr17 | 0 | -1.52 | -1.18 |
| Gli1 | 0 | -1.44 | -1.24 |
| Mob3b | 0 | -1.32 | -1.24 |
| Hoxd1 | 0 | -1.31 | -0.87 |
| Rgs22 | 0 | -1.28 | -0.84 |
| Gdf7 | 0 | -1.27 | -0.99 |
| Cep72 | 0 | -1.2 | -0.75 |
| Vwa3a | 0 | -1.2 | -0.86 |
| Dynlrb2 | 0 | -1.2 | -0.98 |
| Serpinb1 | 0 | -1.19 | -1 |
| Opn4 | 0 | -1.17 | -0.78 |
| Cd180 | 0 | -1.17 | -0.85 |
| Crb1 | 0 | -1.12 | -0.37 |
| Msx1 | 0 | -1.12 | -0.98 |
| Fgfr2 | 0 | -1.11 | -0.87 |
| Dlec1 | 0 | -1.1 | -0.61 |
| Lrrc23 | 0 | -1.08 | -0.78 |
| Myh6 | 0 | -1.06 | -0.78 |
| Pls1 | 0 | -1.05 | -0.73 |
| Neil2 | 0 | -1.05 | -0.29 |
| Calr4 | 0 | -1.04 | -0.79 |
| Efs | 0 | -1.03 | -0.87 |
| Adamts6 | 0 | -1.03 | -0.67 |
| Hhip | 0 | -1.02 | -0.84 |

**D**

| Name | sham | SCI_C | SCI_P |
| --- | --- | --- | --- |
| Gm6408 | 0 | -4.63 | -1.3 |
| Slc26a9 | 0 | -3.9 | 0.009 |
| Olfr1393 | 0 | -3.4 | -0.68 |
| Lrrc27 | 0 | -3.04 | -1.97 |
| Sis | 0 | -2.82 | -1.26 |
| Vmn1r4 | 0 | -2.65 | 0.032 |
| Ctcfl | 0 | -2.54 | 0.263 |
| Klf4 | 0 | -2.47 | -0.77 |
| Esco2 | 0 | -2.45 | -1.34 |
| Olfr545 | 0 | -2.38 | 0.12 |
| Gm47283 | 0 | -2.31 | -0.26 |
| Gm40460 | 0 | -2.31 | -0.71 |
| Wdr86 | 0 | -2.28 | -0.83 |
| Esrp1 | 0 | -2.21 | -0.51 |
| E2f8 | 0 | -2.2 | -1.93 |
| Oard1 | 0 | -2.04 | -2.05 |
| Emilin3 | 0 | -2 | 1.398 |
| Tulp1 | 0 | -1.91 | -0.52 |
| Ccdc153 | 0 | -1.9 | -1.44 |
| Lmntd1 | 0 | -1.89 | -1.28 |
| Iqca | 0 | -1.89 | -1 |
| Pitx1 | 0 | -1.88 | -1.59 |
| Tmem212 | 0 | -1.87 | -1.08 |
| Mc5r | 0 | -1.85 | -1.28 |
| Smim5 | 0 | -1.84 | -1.27 |
| D6Ertd527e | 0 | -1.84 | 0.325 |
| Ccdc146 | 0 | -1.81 | -0.66 |
| Col6a6 | 0 | -1.81 | -0.06 |
| Nme9 | 0 | -1.74 | -1.43 |
| Fam166b | 0 | -1.72 | -1.38 |
| Ildr1 | 0 | -1.71 | -0.99 |
| Adgrf4 | 0 | -1.67 | -0.18 |
| Lgr6 | 0 | -1.66 | -1.49 |
| BC024139 | 0 | -1.65 | -1.22 |
| C1qtnf3 | 0 | -1.62 | -1.38 |
| Acp4 | 0 | -1.61 | 0.21 |
| Dnah8 | 0 | -1.6 | -0.66 |
| Accsl | 0 | -1.6 | 1.376 |
| Stpg1 | 0 | -1.59 | -1.08 |
| Dnah11 | 0 | -1.57 | -0.7 |
| Ninj2 | 0 | -1.57 | -1.23 |
| Slc27a5 | 0 | -1.55 | -0.08 |
| Tex35 | 0 | -1.54 | 0.708 |
| Rad9a | 0 | -1.53 | 1.005 |
| Fscn2 | 0 | -1.52 | 1.093 |
| Ttc16 | 0 | -1.52 | -1.01 |
| Morn5 | 0 | -1.47 | -0.28 |
| Frmpd2 | 0 | -1.47 | -0.64 |
| Col24a1 | 0 | -1.46 | -0.24 |
| Lrit3 | 0 | -1.43 | 0.398 |
| Atp10b | 0 | -1.43 | -1.11 |
| Sctr | 0 | -1.4 | -0.55 |
| Cfap70 | 0 | -1.38 | -0.41 |
| Vwa3b | 0 | -1.38 | -0.59 |
| Cdhr3 | 0 | -1.37 | -0.64 |
| Siglech | 0 | -1.35 | -1 |
| Angptl1 | 0 | -1.34 | -1.49 |
| Kctd14 | 0 | -1.33 | -0.79 |
| Abcg5 | 0 | -1.33 | 1.534 |
| Ing4 | 0 | -1.31 | -0.23 |
| Gm10775 | 0 | -1.29 | -0.07 |
| Mpp4 | 0 | -1.29 | 0.129 |
| Mcm10 | 0 | -1.27 | 0.212 |
| Neu4 | 0 | -1.26 | -1.14 |
| Aipl1 | 0 | -1.25 | 1.265 |
| Olfml1 | 0 | -1.24 | -1.02 |
| Cubn | 0 | -1.24 | -0.12 |
| Barx2 | 0 | -1.23 | -0.31 |
| Slc34a3 | 0 | -1.19 | -0.83 |
| Tmem210 | 0 | -1.17 | -0 |
| Adamts19 | 0 | -1.16 | -0.43 |
| Cavin4 | 0 | -1.16 | -0.7 |
| Col11a1 | 0 | -1.13 | -0.26 |
| H2-Bl | 0 | -1.13 | -0.45 |
| Nudt8 | 0 | -1.12 | -0.32 |
| Chrd | 0 | -1.12 | -0.41 |
| Cfap44 | 0 | -1.12 | -0.1 |
| Casc1 | 0 | -1.1 | -0.24 |
| Gipr | 0 | -1.1 | -0.65 |
| Ccdc162 | 0 | -1.09 | -0.56 |
| Tnfrsf4 | 0 | -1.09 | -0.41 |
| Pcsk4 | 0 | -1.08 | -0.41 |
| Fxyd2 | 0 | -1.05 | -0.57 |
| Kif20b | 0 | -1.05 | -0.58 |
| Pld6 | 0 | -1.04 | -0.44 |
| Entpd4b | 0 | -1.03 | -0 |
| Col11a2 | 0 | -1.03 | -0.47 |
| Riiad1 | 0 | -1.02 | -0.43 |
| Ager | 0 | -1.01 | -0.17 |
| Catsperd | 0 | -1.01 | -0.26 |
| Rxfp1 | 0 | -1.01 | -0.48 |
| Smc1b | 0 | -1 | 0.496 |
| F2rl3 | 0 | -1 | -0.2 |
| Poln | 0 | -1 | -0.24 |

-5  -4  -3  -2  -1  0  1  2
DOWN          UP

**Figure 5 Subcluster analysis of DEGs.** The subclusters of DEGs which have similar expression trends were further analyzed. The log2 (ratios) in SCI_C group ≥1 or ≤−1 was used as a cutoff. Based on log2 (ratios) of the gene expression level relative to that of sham group, the log2 (ratios) of all gene expression levels in sham group were zero. (A and B) The two subclusters which were strongly upregulated following SCI, and then downregulated upon probenecid treatment. (C and D) The two subclusters which were strongly downregulated following SCI, and then upregulated upon probenecid treatment.

Figures 5C and 5D show that the two subclusters were strongly downregulated following SCI and then upregulated upon probenecid treatment. In Fig. 5A, six genes (Cybb, Esam, Itgam, Itgb2, Msn and Ncf2) are demonstrated to have been involved in the leukocyte transendothelial migration signaling pathway; six genes (Col4a1, Erbb2, Flt4, Nos3, Syk and Thbs4) were also involved in the PI3K-Akt signaling pathway. Figure 5B displays three genes (Cyba, Ncf1 and Rac2) involved in the NADPH oxidases, two genes (Cflar and Tnfrsf10b) involved in the TRAIL signaling pathway and eight genes (Cd63, Cyba, Ddx58, Fcer1g, Lyn, Myh9, Ncf1 and Psmb8) involved in the innate immune system. Figures 5C and 5D show that no gene can be clustered into valuable signaling pathways.

## GO enrichment analysis of DEGs

When compared with the sham group, there were 78 GO terms in upregulated DEGs (Fig. 6A; Table S2) and nine GO terms in downregulated DEGs (Fig. 6B; Table S2) in the SCI_C group. The upregulated DEGs were most enriched in: binding, protein binding, chemokine activity, chemokine receptor binding, G-protein coupled receptor binding, anion binding, small GTPase mediated signal transduction, immune system process, immune response. The downregulated DEGs were most enriched in: protein binding, binding, extracellular-glutamate-gated ion channel activity, acid phosphatase activity, transporter activity, mannose metabolic process, excitatory extracellular ligand-gated ion channel activity, transmembrane transporter activity, anion transmembrane and transporter activity. In the SCI_P group, we observed three GO terms in downregulated DEGs (Fig. 6C; Table S3) and no valuable terms in upregulated DEGs (Table S3) compared to the SCI_C group. The downregulated DEGs were protein binding, binding and sequence-specific DNA binding.

## KEGG enrichment analysis of DEGs

Scatter plots were used to express the KEGG enrichment analysis results for the DEGs. When compared to the sham group, the upregulated DEGs in the SCI_C group were most enriched in TNF, NF-kappa B, cytokine–cytokine receptor interaction, Toll-like receptor, Leukocyte transendothelial migration, PI3K-Akt, focal adhesion and apoptosis (Fig. 7A; Table S4); the downregulated DEGs were most enriched in glutamatergic synapse, basal cell carcinoma, axon guidance, other glycan degradation and nicotine addiction (Fig. 7B; Table S4). In the SCI_P group vs. SCI_C group, only the "ECM-receptor interaction" was enriched in the upregulated DEGs (Fig. 7C; Table S5); the downregulated DEGs were enriched in CAMs, malaria, leukocyte transendothelial migration, ECM-receptor interaction, PI3K-Akt signaling pathway, hematopoietic cell lineage, focal adhesion, Rap1 signaling pathway and amebiasis (Fig. 7D; Table S5).

## DISCUSSION

Recent studies have shown that probenecid has neuroprotective and repairing effects on brain disorders (Wei et al., 2015; Xiong et al., 2014). However, its effect on genome-wide transcription in SCI is still unknown. Therefore, in this study, RNA-Seq was used to analyze the effect of probenecid on the local expression of gene transcription 8 h after SCI.

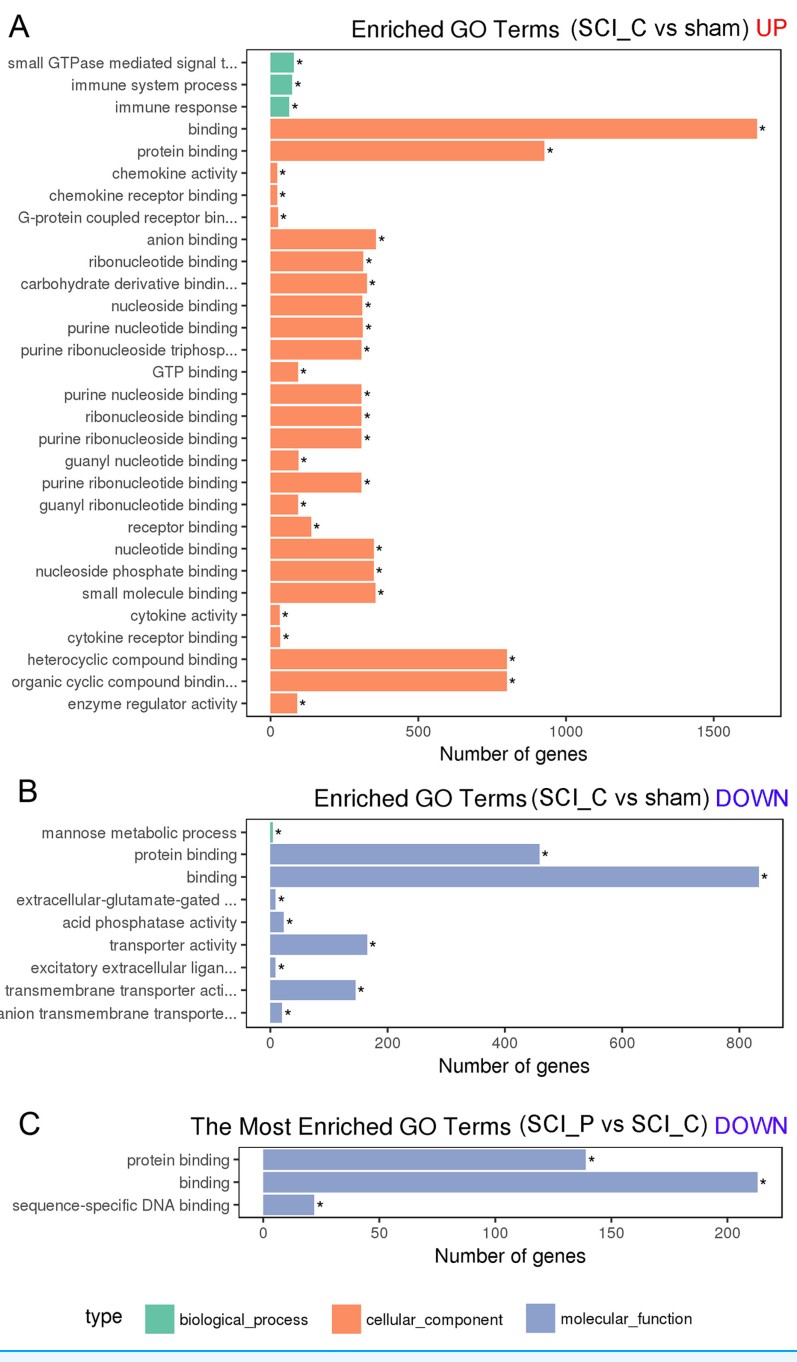

**Figure 6 GO enrichment analysis of DEGs.** DEGs were implemented by the GOseq R package, in which gene length bias was corrected. GO terms with corrected *P* value ≤ 0.05 were considered significantly enriched by DEGs. The asterisk (*) represent significant enrichment terms (*P* ≤ 0.05). (A) GO analysis of upregulated DEGs in the SCI_C vs. sham group. (B) GO analysis of downregulated DEGs in the SCI_C vs. sham group. (C) GO analysis of downregulated DEGs in the SCI_P vs. SCI_C group.

The results showed that when compared with the sham group, there were 4,617 DEGs in the SCI_C group, including 2,904 upregulated and 1,713 downregulated genes. When compared with the SCI_C group, there were 641 DEGs in the SCI_P group, 286 of which

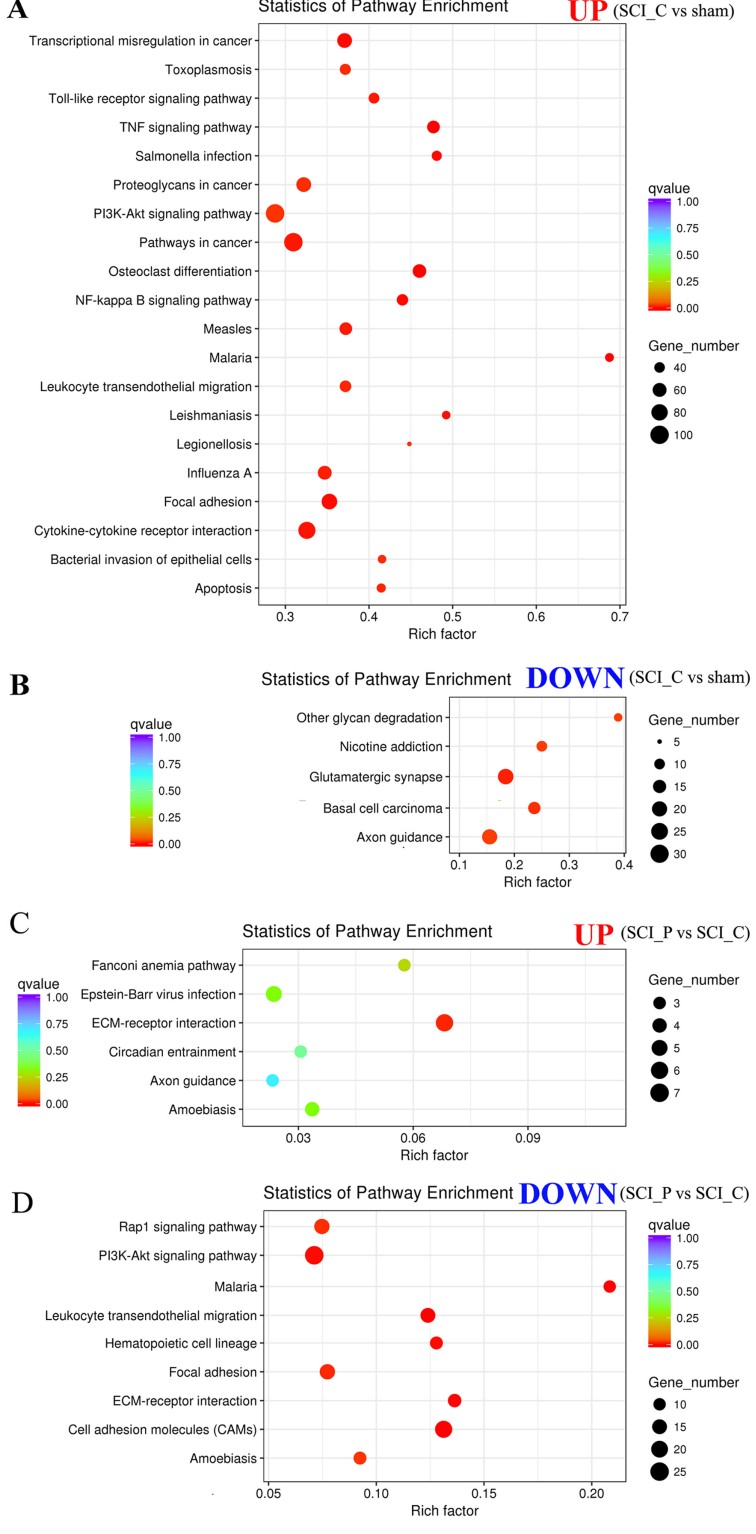

**Figure 7 KEGG enrichment analyses of DEGs.** KOBAS software was used to test the statistical enrichment of DEGs in KEGG pathways. In this figure, KEGG enrichment is measured by Rich factor, *Q* value and the number of genes enriched in the related pathway. Rich factor refers to the ratio of the number of differentiated genes (sample number) enriched in the pathway to the number of annotated genes (background number). The larger the Rich factor, the greater the degree of enrichment. *Q* value is

**Figure 7** (continued)
the *P* value after multiple hypothesis test correction. The range of *Q* value is between 0 and 1. The closer the *Q* value is to 0, the more significant the enrichment is. The KEGG pathways were shown in (A) upregulated DEGs (SCI_C vs. sham); (B) downregulated DEGs; (C) upregulated DEGs (SCI_P vs. SCP_C); (D) downregulated DEGs (SCI_C vs. sham).

were upregulated and 355 downregulated. These are consistent with others' and our previous reports (*Chen et al., 2013*; *Shi et al., 2017*). It also shows that the results of this experiment are reliable. As compared to the SCI_C, there were 641 DEGs in the SCI_P group, 286 were upregulated and 355 were downregulated. To further verify the RNA-seq results, we randomly selected nine DEGs (Itga1, Lamb1, Cldn5, Lama2, CD34, Esam, Setdb2, Agrn and Ccnt2) for RT-qPCR. The results showed that the expression patterns of these genes detected by these two methods were similar, indicating that our RNA-seq results are reliable and can be used for subsequent analysis. These also confirmed that probenecid can alter gene transcription after SCI.

To further analyze the DEGs effected by probenecid, we used GO enrichment to reflect the distribution of DEGs on GO term enriched in cell components, molecular functions and biological processes (*Huang et al., 2013*). In the SCI_P vs. SCI_C group, analysis showed that there were three GO terms in downregulated DEGs (protein binding, binding and sequence-specific DNA binding) and no valuable terms in upregulated DEGs. KEGG analysis showed that the valuable signaling pathways associated with these DEGs included: CAMs, leukocyte transendothelial migration, ECM-receptor interaction, PI3K-Akt signaling pathway, hematopoietic cell lineage, focal adhesion and Rap1 signaling pathway.

Among these signal pathways, some are known to be related to SCI, such as CAMs (*Brook et al., 2000*; *Zhang et al., 2008*), ECM-receptor interaction (*Zhou et al., 2017*), PI3K-Akt signaling pathway (*Li et al., 2019a*, *2019b*; *Zhang et al., 2017*) and focal adhesion (*Chuang et al., 2018*; *Graham et al., 2016*; *Hao et al., 2018*).

Following SCI, probenecid treatment could downregulate some genes, subclusters and signaling pathways. Leukocyte transendothelial migration from the blood into tissues is vital for immune surveillance and inflammation (*Cook-Mills, 2006*). However there is a large amount of leukocyte infiltration involved in the pathological process of SCI. These infiltrating leukocytes need to bind to endothelial CAM and then migrate between vascular endothelial cells (*Wang et al., 2011*). Therefore, the inhibition of leukocyte transendothelial migration and CAMs induced by probenecid may play a role in inhibiting inflammation by weakening the infiltration of white blood cells in the injured area. In this study, we clustered six genes (Cybb, Esam, Itgam, Itgb2, Msn and Ncf2) involved in this pathway. Their expression is strongly downregulated following SCI and then upregulated upon probenecid treatment. This demonstrates that probenecid treatment following SCI can play an anti-inflammatory role by inhibiting the infiltration of inflammatory cells.

The ECM plays an important role in tissue and organ morphogenesis (*Bonnans, Chou & Werb, 2014*; *Rabelink et al., 2017*) and control of cellular activities such as adhesion, migration, differentiation, proliferation and apoptosis (*Yue, 2014*). Focal

adhesions are specialized, intracellular sites in which aggregated integrin receptors interact with extracellular matrices, while extracellular matrices interact with intracellular actin cytoskeleton (*Burridge, 2017*; *LaFlamme et al., 2018*). At the same time, focal adhesions are the result of cell and extracellular matrix (ECM) interactions (*Burridge, 2017*; *De Pascalis & Etienne-Manneville, 2017*). ECM and focal adhesions are downregulated after probenecid treatment, indicating that probenecid might improve SCI by inhibiting adhesion, migration, differentiation, proliferation and apoptosis.

It has been reported that PI3K-Akt signaling fuzes a variety of extracellular and intracellular signal transduction pathways that regulate macrophage biology, including: the production of pro-inflammatory cytokines, phagocytosis, autophagy and homeostasis (*Vergadi et al., 2017*). PI3K-Akt signal pathway is downregulated in SCI after probenecid treatment, and the six genes (Col4a1, Erbb2, Flt4, Nos3, Syk and Thbs4) being clustered into this pathway indicate that probenecid might improve SCI by regulating macrophages and inhibiting inflammatory pathways. This is likely to provide important clues towards identifying the mechanism by which probenecid acts.

The relationship between the hematopoietic cell lineage pathway and SCI was referred to in a report on the bioinformatics analysis of SCI (*Zhu et al., 2017*). Its specific role has yet to be reported and requires further discussion.

Rap1 signal pathway plays an important role in regulating cell–cell and cell–matrix interactions by regulating the function of adhesion molecules (*Kim, Ye & Ginsberg, 2011*; *Pollan et al., 2018*). In our study, Rap1 signaling pathways were enriched in downregulated DEGs of SCI after probenecid treatment, suggesting that probenecid may inhibit cell adhesion and polarization by inhibiting the Rap1 signaling pathway, thereby inhibiting inflammation.

In addition, three genes (Cyba, Ncf1 and Rac2) related to the NADPH oxidases, two genes (Cflar and Tnfrsf10b) related to the TRAIL signaling pathway and eight genes (Cd63, Cyba, Ddx58, Fcer1g, Lyn, Myh9, Ncf1 and Psmb8) related to the innate immune system were also strongly downregulated following probenecid treatment. We know that NADPH oxidases are involved in oxidative stress, the TRALL signaling pathway mediates inflammation and apoptosis and the immune system is involved in almost all pathological processes of injury (*Chyuan et al., 2018*; *Ewald, 2018*; *Tisato et al., 2018*). Therefore, probenecid treatment can potentially play a neuroprotective role by inhibiting immune response, oxidative stress, anti-inflammation and anti-apoptosis after SCI.

## CONCLUSIONS

Acute SCI can lead to changes in the mRNAs of injured spinal cords. These mRNAs and their related pathways can provide some explanation for the pathological mechanism of acute SCI. More interestingly, we also demonstrated that probenecid can lead to gene expression inhibitions in an acute injured spinal cord. These downregulated DEGs and their associated signaling pathways—such as focal adhesions, leukocyte transendothelial migration, ECM-receptor interaction, PI3K-Akt, Rap1—are mainly related to inflammatory response, local hypoxia, macrophage differentiation, adhesion migration and apoptosis of local cells. This suggests that the application of probenecid in the acute

phase can improve the local microenvironment of SCI. However, the application of probenecid as a therapeutic drug for SCI requires further investigation. Next, the detailed research on this subject will be conducted by combining animal models and clinical practice.

### Funding

This work was supported by grants from the National Natural Science Foundation of China (Nos. 81571194 and 81772321). The funders had no role in study design, data collection and analysis, decision to publish, or preparation of the manuscript.

### Grant Disclosures

The following grant information was disclosed by the authors:
National Natural Science Foundation of China: 81571194 and 81772321.

### Competing Interests

The authors declare that they have no competing interests.

### Author Contributions

- Yu-Xin Zhang performed the experiments, analyzed the data, prepared figures and/or tables, and approved the final draft.
- Sai-Nan Wang performed the experiments, analyzed the data, prepared figures and/or tables, and approved the final draft.
- Jing Chen performed the experiments, prepared figures and/or tables, and approved the final draft.
- Jian-Guo Hu conceived and designed the experiments, authored or reviewed drafts of the paper, and approved the final draft.
- He-Zuo Lü conceived and designed the experiments, prepared figures and/or tables, authored or reviewed drafts of the paper, and approved the final draft.

### Animal Ethics

The following information was supplied relating to ethical approvals (i.e., approving body and any reference numbers):

The Animal Care and Use Committee of Bengbu Medical College provided full approval for this research (037/2017). Animal care following surgery was carried out in compliance with the regulations for the management of experimental animals (revised by the Ministry of Science and Technology of China in June 2004).

### Data Availability

Data is available at NCBI PRJNA554464: SAMN12268369, SAMN12268370, SAMN12268371, SAMN12268372, SAMN12268373, SAMN12268374, SAMN12268363, SAMN12268364 and SAMN12268365.
## Supplemental Information

Supplemental information for this article can be found online at http://dx.doi.org/10.7717/peerj.8367#supplemental-information.

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
