# Peer review of "A transcriptomic study of probenecid on injured spinal cords in mice"

_PeerJ, doi:10.7717/peerj.8367_

## Round 0.1 · original submission · Major Revisions

Your manuscript has been reviewed and the reviewers have a number of concerns. In particular, adequate statistical analysis should be performed according to the suggestions of the reviewers.

In addition, the English should be carefully revised.

Reviewer 1 ·

Basic reporting

In this work Yu-Xin Zhang and co-workers investigate the effect of probenecid in local gene expression, upon spinal cord injury in mice. Probenecid has been shown to have a neuroprotective effect in some brain disorders; however, its role in modulating the transcriptional response upon spinal cord injury (SCI) had not been addressed yet. To answer this question the authors performed an RNA-seq analysis in spinal cord samples derived from un-lesioned and lesioned mice, treated or not with 0.5 mg/kg of probenecid immediately after injury. Samples were harvested 8 hours after the lesion, to investigate primary changes in gene expression.

The paper is self-contained and addresses a well-defined question with the appropriate methodology. Relevant literature is cited and the results obtained are properly discussed within the state-of-the-art in the field. Maybe some more information could be extracted (and discussed) from the data (see point 4 below).

Raw data are available in a public repository as requested

Experimental design

The authors address a relevant, well-defined question that fills an important gap of knowledge regarding the transcriptional response observed during central nervous system injuries. Overall, the work is technically well done, however, several experimental and reporting caveats should be addressed before publication:

1) The number of biological replicates for RNA-seq experiments on animals is very low (n=3). This is also reflected in the PCA analysis, where probenecid samples do not cluster together and they do not separate well from control-treated samples. The authors should consider increasing the number of treated animals to improve differential gene expression analysis and data consistency.

2) Statistical analysis for the qRT-PCR experiments in figure 3 is missing. Was the analysis performed in independent samples or in the same samples used for the RNA-seq analysis? Are the differences observed statistically significant? This info needs to be provided prior to publication. Figure legend should be also re-written to improve clarity.

3) GO and KEGG analysis on figures 5, 6 and 7 should include only enriched categories that are statistically significant. Many of the pathways represented in the figures are not significative and therefore should be eliminated to avoid confusion and to strengthen the manuscript conclusions.

4) In figure 4 the authors describe the hierarchical clusterization of the RNA-seq data. However, they do not give any indication on the genes (or gene categories) within each of these clusters. For instance, there is a cluster that is strongly downregulated in SCI_control versus SCI_sham, and upregulated again upon probenecid treatment. Are those genes related to neuroprotection? Such information is likely to provide important clues into the mechanism of action of probenecid and should be addressed in the manuscript. In addition, it is not clear the criteria used for the clusterization. Does the analysis include all DEG between the three conditions? Or is it based in the DEGs with the lowest p value? The authors should clarify these aspects.

5) In general, the description of the experimental approach (in particular regarding the bioinformatics analysis) and figures labelling and legends do not give sufficient detail, making it difficult to properly interpret the results at several points.

Validity of the findings

This manuscript provides a relevant dataset that could be used in future work to elucidate the mechanism of action of probenecid in SCI models. In addition they are properly discussed, with relevant references to available literature and no “over-stating” conclusions.

Reviewer 2 ·

Basic reporting

Some typos in verbs and syntax improvements possible and the quality of the figures could be improved. Some of them are illegible (for instance Fig. 5, Fig. 6 and Fig. 7). Raw data are supplied.

Experimental design

Although the research question is well defined and it is of potential interest, the conclusions remain elusive. The effect of the drug used by the authors (probenecid) on the general gene expression is convincing, but if the drug exerts neuroprotective effects on the injured spinal cord and through which genes and/or signaling pathway/s are not investigated at all.
Anyway, methods are well-described and present sufficient details.

Validity of the findings

The conclusions are not supported by the results, but may only represent speculations. Although the research question of the authors is interesting, the manuscript in its current form does not have sufficient novelty to be published.

Additional comments

In the present manuscript Zhang and colleagues evaluated the effect of the probenecid on general gene expression in spinal cords following acute spinal cord injury (SCI). Briefly, the authors performed an RNA Seq analysis in control and SCI mice with or not the probenecid treatment.
They identified 641 differentially expressed genes in probenecid-treated SCI mice compared to vehicle-treated SCI mice, of which 286 were up-regulated and 355 were down-regulated. To validate the genome wide analysis, the authors evaluated the relative expression level of only nine genes, which represent only less than 2%. Furthermore the analysis lacks of a statistical analysis.
Finally the authors performed a Go and KEGG enrichment analysis to highlight the functional categories and signaling pathway/s, respectively, to which differentially expressed genes belong. The down-regulated genes in probenecid-treated SCI mice clustered in three GO terms (protein binding, binding and sequence-specific DNA binding), although none of the up-regulated genes was significantly enriched in Go terms. Some of the genes, resulting differentially regulated by probenecid treatment in SCI mice, belonged to some SCI-related signaling pathways (CAMs, ECM-receptor interaction, PI3K-Akt signaling pathway and Focal adhesion), as evidenced by KEGG enrichment analysis.
Although these data are of potential interest, Go and KEGG enrichment analysis can represent only the initial step of a work to better address the experiments that must be done to support and validate a scientific hypothesis, but certainly not the conclusion of a scientific report.
Furthermore, the experiments performed in this study are very preliminary and the most of conclusions of the authors represent only possible hypotheses. Moreover, an important issue that the authors did not consider is whether the treatment of probenecid in their system had “neuroprotective and repairing effects” in injured spinal cord. This point, in my opinion, represents the major weakness of the manuscript.
The manuscript is really poorly written and makes it difficult to understand the text in some points. For all these reason I believe that this manuscript should not be committed for the publication in Peer Journal.

Reviewer 3 ·

Basic reporting

I suggest a small improvement in background citations (see comments section)

Experimental design

Spinal Cord Injury (SCI) is a devastating clinical condition resulting mainly from traumatic damage that
leads to neuronal cell death and concomitantly, different degrees of motor function loss and sensitivity.
Neuroprotective treatment is one of the promising strategies to alleviate SCI symptom and increase the successful of others pharmacological or rehabilitating strategies. Probenecid is a urate transporter-1 (URAT-1) inhibitor, that has been recently demonstrate to exert a neuroprotective and repairing effects in some brain disorder.
In this manuscript Zhang and colleagues analyze and make available an RNA-seq dataset of spinal cord tissue derived from control, SCI and Probenecid treated SCI mice.
The manuscript is clear and well written and the raw sequence data have been deposited into Sequence Read Archive. Zhang and colleagues analyzed 3 replicates for each experimental point (CTR, SCI and SCI+ Probenecid), using a classical RNA-seq approach. The experimental methods are clear and well explained with sufficient amount of information.

Validity of the findings

A basic but well performed analysis of SCI and Probenecid treated SCI compare to control tissue was presented. The conclusion of the manuscript is supported by the performed analyses. Although the authors have not entered in deep into the merits of the mechanism of action of the drug, I believe that the published dataset is an excellent starting point for other studies and a valid tool for designing new pharmacological strategies in the treatment of spinal contusion. Indeed, authors identify macrophages and more in general the inflammatory response as a primary target of Probenecid treatment, speculating that this drug can exert some beneficial effects in SCI therapy.

Additional comments

From my point of view, I think that the publication in this journal is appropriate.
Nevertheless, I would suggest the following minor changes to the authors in order to improve the manuscript:
-Line43: correct “rats” with “mice”
-Line81-90: background can be expanded to other studies involving Probenecid and nerve injury. Some example are listed bilow:
a) Probenecid and N-Acetylcysteine Prevent Loss of Intracellular Glutathione and Inhibit Neuronal Death after Mechanical Stretch Injury In Vitro. Du L, Empey PE, Ji J, Chao H, Kochanek PM, Bayır H, Clark RS. J Neurotrauma. 2016 Oct 15;33(20):1913-1917.

b) Pannexin 1: a novel participant in neuropathic pain signaling in the rat spinal cord. Bravo D, Ibarra P, Retamal J, Pelissier T, Laurido C, Hernandez A, Constandil L. Pain. 2014 Oct;155(10):2108-15.

c) The L-kynurenine-probenecid combination reduces neuropathic pain in rats. Pineda-Farias JB, Pérez-Severiano F, González-Esquivel DF, Barragán-Iglesias P, Bravo-Hernández M, Cervantes-Durán C, Aguilera P, Ríos C, Granados-Soto V. Eur J Pain. 2013 Oct;17(9):1365-73.

-Line: 126-127: specify if a cut-off was used or not for fold change.
-Line146: a comment on the result depicted in the PCA would be useful (distance between CTR and SCI etc ..)
-Line 148: I would change the title of the paragraph with: “The effect of SCI and Probenecid treatment on gene expression”

-Line162/Fig3: I would suggest to shows readcounts replicates in the graph.
-Table 1: there is an error (probably derived from a cut and paste) in the sequence of Itga1 primers (not match)

---

## Round 0.2 · Minor Revisions

In accordance with the Reviewers comments, a revision of the English is still strongly recommended. Although PeerJ does provide a Language Editing Service you are at liberty to use any service or English speaking colleagues etc.

Reviewer 1 ·

Basic reporting

The authors have now address all my concerns. I consider the article suitable for publication, but I suggest it undergoes PeerJ English language editing services to improve the clarity of the message.

Experimental design

See above

Validity of the findings

See above

Additional comments

See above

Reviewer 2 ·

Basic reporting

see comments for the author below

Experimental design

see comments for the author below

Validity of the findings

see comments for the author below

Additional comments

In the revised version of the manuscript the authors have sufficiently improved the quality of the figures and the English language. Anyway, we strongly recommend using the language editing services provided by PeerJ journal.

---

## Round 0.3 · accepted · Accept

In the present revised version all criticisms have been adequately amended.